# The Sonic Intra-Face of a Noisy Feminist Social Kitchen

**Juliana España Keller** 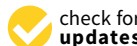

Victorian College of the Arts, University of Melbourne, Melbourne, VIC 3006, Australia;
info@julianaespanakeller.com

**Abstract:** This paper asks what is the value of transforming the kitchen into a sonic performative work and public site for art and social practice. A Public Kitchen is formed by recreating the private and domestic space of a kitchen into a public space through a sonic performance artwork. The kitchen table is a platform for exploring, repositioning and amplifying kitchen tools as material phenomena through electronic and manual manipulation into an immersive sonic performance installation. This platform becomes a collaborative social space, where somatic movement and sensory, sonic power of the repositioned kitchen tools are built on a relational architecture of iterative sound performances that position the art historical and the sociopolitical, transforming disciplinary interpretations of the body and technology as something that is not specifically exclusively human but post-human. A Public Kitchen represents a pedagogical strategy for organizing and responding collectively to the local, operating as an independent nomadic event that speaks through a creative practice that is an unfolding process. (Re)imagining the social in a Public Kitchen produces noisy affects in a sonic intra-face that can contribute to transforming our social imaginations, forming daring dissonant narratives that feed post-human ethical practices and feminist genealogies. This paper reveals what matters—a feminist struggle invaluable in channeling the intra-personal; through the entanglement of the self, where language, meaning and subjectivity are relational to human difference and to what is felt from the social, what informs from a multi-cultural nomadic existence and diffractive perspective. The labored body is entangled with post-human contingencies of food preparation, family and social history, ritual, tradition, social geography, local politics, and women's oppression; and is resonant and communicates as a site where new sonic techniques of existence are created and experiences shared.

**Keywords:** sound and noise art; feminist new materialism; posthumanities; doing-cooking; social engagement; participatory practices

## Preface

This paper needs to expand out from the page for the reader. Firstly, and most obviously, it offers a written analysis of the artwork and emphasizes the relation to Martha Rosler's seminal work, *Semiotics of the Kitchen* (Rosler 1975). A Public Kitchen responds to the historical, socio-political, feminist academic research in relation to the Special Issue on Feminist New Materialisms. Secondly, visual images of various performance iterations provide useful illustration. Finally, this text offers a link to the sound performance work, 'The Kitchen Shift' that can be accessed online. This audio-visual documentation can be found on my website with the following hyperlink listed below. By using this link, the reader is connected to the art work directly. This adjunct helps elucidate a fuller understanding of a Public Kitchen.

This journal paper forms part of the artwork and research, *A Public Kitchen*, which is foremost a collective performance work and immersive sound installation. When reading this text, there is a desire and need (by the artist) for the reader to experience the collective sensory and sonic attributes

of a Public Kitchen; however, the live and immersive experience of performance art, or live art, is at once irreplaceable or not easily substituted. The artwork is thus always to be set-up each time for an indeterminate and distinct outcome with room for failure and noisy slippages. A Public Kitchen is a culturally constructed representation of reality and positions what is possible and what it is not possible in its activation. As there have been many iterations of this work to date—spanning geographies and social groups (2015–2019)—much vital material has been generated over time in the form of audio-visual documentation.[1]

## 1. Introduction

> "dare take the risk of affirmative politics and the collective construction of social horizons of hope" (Rosi Braidotti 2014).

The research generated by the artwork, a Public Kitchen, contributes to feminist new materialist discourse by making the notion of human–non-human agency graspable as a creative act that cuts across pre-established dichotomies, by transversing hierarchies of power relations that organize human life as contextualized in by new materialist theorists Barad and Braidotti (Barad 2003, 2007; Braidotti 2002, 2013). Through the medium of sound, I am investigating; emphasizing entities, ethics, social class, and social political intervention in the process. This subjectivity illuminates and activates how we move through the world, react to surroundings and respond to everything. It also shows how the normative and hierarchical relations amongst human groups based on race, sexuality, social class, and ability are always intimately entangled with the broader political economies/ecologies of which we are a part. The performance artwork establishes a scaffold for thinking about a range of ideas of what is felt through encounters with philosophy, sonic arts, community participation, feminist materialism, and post-human thought.

The human and non-human relation to machines and machine learning is enacted through intra-active entanglement; since it represents an active pedagogy practice for organizing, facilitating and responding collectively to the local activated as Andrew Murphie states on how to anarchive[2] through: "the ability to find a way out of systems, often from within to life's living"(Murphie 2016, p. 5). A noisy kitchen is felt as a musical sounding in the everyday rhythm of lived intensities and is seen as Brian Massumi (2002) posits "*a repertory of traces* of collaborative research-creation events" and "platforms for organizing and orienting live, collaborative *encounters*" (Murphie 2016, p. 6). Agency is conjured through the doing-cooking[3] of the kitchen to create a sonic recipe[4] as the becoming of the human–non-human relationship to uncover the paradigms that shape-shift performance art.

This paper focuses on the unpacking of a creative practice, resonating with agency and amplifying where the experiential is intensely experienced as "creating together", foregrounding a way to move forward pedagogically and experientially (Conrad and Sinner 2015; Manning 2016). A Public Kitchen is embedded in the social, and the artwork allows for human slippages, failures that form part of the work. In particular, it considers how feminist new materialism can create daring dissonant narratives that feed post-human ethical practices and feminist genealogies. This research reveals what matters—a feminist struggle invaluable in highlighting and responding collectively to the local with a systemic understanding of material phenomena in an immersive sonic performative installation. This

---

[1] The creative artwork and exhibition titled: "The Kitchen Shift" can be found here: https://vimeo.com/332604174. The Password: THEKITCHENSHIFT OR on the artist's website: http://cargocollective.com/julianaespanakeller/THE-KITCHEN-SHIFT-THE-INDUSTRIAL-SCHOOL-BUILDING.

[2] The word 'anarchive' comes from the "The Go-To How to Book of Anarchiving" by Andrew Murphie of the Senselab (Murphie 2016).

[3] The entanglement of "doing-cooking"—a term coined by Michel de Certeau (de Certeau et al. 2014).

[4] *A sonic recipe* is a dynamic arrangement of sound/noise material generated through the artwork by the intersection of the human–non-human, embracing the critical potentiality of a vital matter in a sonic apparatus.

position seeks to ignite and transform our social imagination and deactivate pervasive and dominating patriarchal ethico-politics.

Finally, this paper explores ways in which post-humanist and new materialist thinking can be put to work in order to (re)imagine a more open perspective in approaching and pursuing community-based, collaborative practices underscored pedagogically and rhizomatically as a teaching about gender and a politics of care (Revelles-Benavente and Ramos 2017; Kumar 2002). What performance practices can do is contribute, co-produce through a constellation of variables of agency and become activated in a collaborative learning experience (Strom and Martin 2017). It is an invitation to scholars of posthumanism and new materialism to imagine how creative ideas and processual thinking might be put to work through performative practices with fluidity, flux, expansion and understanding of difference. In observation, anarchiving pertains to the event and live art that is activated through a performative cross-platform phenomenon where humanity is something that needs more humanizing as we move forward to challenging times ahead, where participation is risky and where research-creation can be contextualized performatively as co-constructed. In this way, human–non-humans, actions, or events are defined by their relations and function as part of an assemblage[5] that is concerned with processual work and self-transformation. (Huybrechts 2014; Torrens 2014). What is correlative to intra-corporeality—where the artwork is an aesthetic, psychological and physical experience—is how these entanglements mesh with our perceptions; where the mediation of affect as a sound performance plays out and functions as a reading of the bodies of others.

## 2. Starting from the Middle

*Eight individuals in identical dresses file robotically in through a portal door. They walk, one after the next, into a space that is embedded with history, bringing their ways of moving materiality into the present. Heavy, heavier, heaviest. They walk in formation, masks on, heads high, hands up, carried above their chest, ready to work in thick, yellow, rubber utilitarian gloves. Each individual picks a spot in the room and stares out at the audience with a dead-pan face. One of the members picks up her guitar and begins to play doom-drone riffs. The sound does not rock, as much as crush[6] inexorably, until the rest of the group move out collectively into the performative space with their bodies and making their way over to the kitchen table—the motherboard—to join her. Their self/bodies move through the space, shifting into various somatic positions on the floor, against the wall, with each other. The figure/ground choreographic relation between the human and this historical place dissolves as the outline of the human is traversed by substantial material intra-changes; the prelude of oneself as transcendent, generated through and entangled with other systems, processes and events. Their shiny bodies cannot resist the allure of shiny objects waiting on the motherboard, considering the effects they have, from manufacture to disposability, while reckoning with the strange agencies that intra-connect substance, flesh and place on a deep molecular level. Vibrating, vibrating . . . the sound distils and subtracts, and then the vocals begin resonating, looping in and generating live outbursts from the artist who is writing these words now, plunging the sound into a human-non-human realm of culinary noise abstraction. Rubber gloves smack back at you. The sonic recipe is in effect.*

Here, I begin by writing to you from the middle of a Public Kitchen, as the words are moving across and through the whole final creative artwork. The middle is for me a happening of writing and research that emerges from my own personal experience combined with an art practice that requires a sensitivity, or attunement, to what moves through thought and thinking as a performance artist—a

---

[5]   "Assemblage" is a material affective dynamic in which bodies become "other" to themselves (Deleuze et al. 1987). However, my understanding moves further in the direction of post-humanist feminist epistemology and quantum physics, as outlined in the works of Donna Haraway and Karen Barad. In turn, difference is relational to the concept of "becoming" explained by Barad and Braidotti. In this paper, I am articulating that participating with other bodies in a Public Kitchen is thus a learning process and can be used as a pedagogical tool to understand others.
[6]   The word "crush" describes noise as "coming on," or putting sonic pressure on something.

maker of live art.[7] I am creating a relation to the person who is reading this paper, by moving into a thinking process that (re)constructs our interdependence with the human and nonhuman, beginning with thinking and acting diffractively[8] and intra-actively[9] without discrediting the consideration for collectivity and *the other*[10] and all the challenges of foregrounding this artwork. A Public Kitchen creates an opening, unfolding as a performative choreography; it is a creative event that communicates, acts and behaves in a social space, all the way down to the molecular level, not too rigidly conformist, but a writing canvas for the one, or the few, who speak to many. She rises.

A *Public Kitchen* is defined as a sonic performance with innumerable potential iterations as a technique for making research-creation across geographical locations, made possible via an immersive sculptural apparatus and active human participation. The participants in this social artwork are women, or those who identify as women, inclusive of gender fluidity. In turn, the objects generate a sonic performance work that dramatizes the material agencies, flows and intensities between kitchen objects and humans. This is done by creating sonic recipes in a Public Kitchen installation space. Living always comes to terms with forms of dissonance emerging from a complex set of social conditions, such as the auditory experience of sound that lacks musical quality; the sound that is a disagreeable auditory experience as a form of noise. Sound matter is therefore generated by making these conditions/forces a constant process of engagement, where thinking and acting move 'from the middle out'—drawing on what is intrinsic or embedded, creating ways of shifting into each other and attuning to these fields of difference. A Public Kitchen can, therefore, be seen to represent an active pedagogy for organizing and responding collectively to the local, through a spectrum of sound phenomena where home is a middling, while still operating as an independent nomadic event with many, and potentially on-going, transnational iterations.

I argue that human–non-human intra-actions within sonic recipes are made apparent due to the doing of the action by the subject, paraphrasing Barad, being the context of the work; this makes the effects of the action relational in the real world (Barad 2003). Intra-action should not be confused with interaction, where elements exist first and then interact. Instead, an intra-action conceptualizes that it is the action between (and not in-between) that which matters; it is in the action that the elements themselves are produced interdependently. In turn, a sonic recipe (in which intra-actions can occur) is defined as a relational partition of electronic sound patterns that form a dynamic arrangement of sound/noise material composed with kitchen tools and appliances in a sonic performance installation (a Public Kitchen). Overall, sound phenomena materialize when the human and non-human are fully present and exercise a guiding or piloting role, affecting and working through what is organically presented in a particular iteration of a Public Kitchen event. Here, the idea of the participatory is critical, where the human and non-human fuse to take on a positive and affirmative character, aligning with Braidotti's understanding of an ethical practice, which is an ethico-politic stated in the opening quote of the introduction. I would also posit it as, what Braidotti distinguishes, "a break with the doxa": the acquiescent application of established norms and values by de-territorialising them and introducing an alternative ethic flows (Braidotti and Hlavajova 2018, p. 224).

I contend that a participatory sonic performance work precipitates a kitchen interior intra-subjectivity—a process evoking the specific entanglement of *doing-cooking* with *affect* by providing a complex assemblage of embedded and embodied thought, where the psyche and the body work in

---

[7] This thinking has been influenced immensely by Erin Manning, who conceptualizes "middling" as a way of thinking through the way the minor (a minor gesture that moves across the work) calls us to attend to something and moves us through thresholds of socialities and techniques in philosophy, art and activism (Manning 2019).

[8] (Barad 2007).

[9] (Barad 2003).

[10] In this journal paper, I am articulating that participating with other bodies in a Public Kitchen is thus a learning process and can be used as a pedagogical tool to understand others. I refer here to American philosopher and feminist scholar, Bell Hooks, and the intersectionality of race, capitalism, and gender, and what she describes as the ability to counter-act systems of women's oppression and class domination (Hooks 2000).

unison. What is correlative to intra-corporeality—where the artwork is an aesthetic, psychological and physical experience—is how these entanglements mesh with our perceptions; where the mediation of affect as a sound performance plays out and functions as a reading of the bodies of others. Within each iteration, each participant is triggered by the vibrational sensation of sound that rises to the surface through the mind/body. Affect can be felt as sound that behaves as active matter—to listen and absorb sound material activated through the playing of kitchen objects, deeply. This triggers physical movement felt in the transmission within and between bodies and objects.

## 3. Resonance in a Public Kitchen

Resonance in a Public kitchen addresses a deeply intertwined collective moment when the human and non-human intra-act in the apparatus of a Public Kitchen. To be clear, resonance is one of the fundamental phenomena, not just of acoustics or science in general. Resonance is a factor entangled with intra-activity and diffraction in the sonic intra-face of a Public Kitchen. The "resonance" experienced becomes a vital and transformative moment in the process, instigated by a "material turn". A critical live awareness occurs where sound (technological or non-technological) can be felt and seen collectively and affectively as an automated society coping with the demands of life with everyday technology. Thus, resonance in performance can be seen as a bridge to the post-human: to be attentive to the mutual accommodation or responsiveness of human and non-human agents. It is not mitigated from the outside to the inside of the body; it is already in the body and in the mind through active resonant forces. In turn, my position always returns to the sound that is produced in intra-activity. Within each iteration, each participant is triggered by the vibrational sensation of sound that rises to the surface through the mind/body with the gestural use of a contac mic that touches each kitchen tool as it is played. Affect can be felt as *sound that behaves as active matter*—to listen and absorb sound material activated through the playing of kitchen objects, deeply. This triggers physical movement felt in the transmission within and between bodies.

In turn, this research seeks to address resonant frequency characterized in the oscillation of sound, which can be observed as vibrational bodies performing with the kitchen objects (reverberating as an extension of the self). I argue that it must always be understood relationally, as vibration that is already materializing between humans and non-humans in intra-action. This also raises an awareness of the complexities involved with diffractive paradoxes of difference revealed in sonic relations and my own affective politics. In turn, sound works its way to the forefront of contemporary sensory behaviour in user experience, by sculpting, shifting and changing our perception of the kitchen environment in which the body labors to listen creatively. I argue that a Public Kitchen becomes an echo chamber of kitchen intensities that resonate amongst others, pushing thought toward its material forces of intra-actions to describe as Manning states, "pure experience in the in-folding of potential that keeps actual experience open to its more than" ([Manning 2016](), p. 29).

As Elaine Swan considers, there are colonial and anti-colonial dynamics at play between masculinity and femininity, specific ethnicities, multiculturalism, and imperialism, as I have explained in my footnote on "the other". I argue that these are evident in a Public Kitchen assemblage. I would also argue that these dynamics are seen as post-human indicators from which we can interrogate, more closely, the connection (imagined or not) between food and the Other. As a multicultural pedagogy that can perform and engender, this research and artwork certainly experiments with various encounters with 'otherness'; this is approached via a dialogical response to feminism, social class and domestic labor. In post-human terms, food has, as Swan states, "become a battleground for politics, policy, and reform" on many levels ([Flowers and Swan 2011](), p. 235).

In the context of transnational mobility, Ilaria Vanni attributes the sense of "being at home," or belonging to somewhere, as dissociated from a geographical location and replaced by belonging through specific everyday practices ([Lloyd and Vasta 2017]()). This moving or roving idea is conceptualized in the Artists' Soup Kitchen, based in Toronto, Canada, where different artists host a lunch each week and bring their creative practices to a community table. Vanni makes us aware that the concept of home

has been an intellectual obsession for a long time; where others try to pin down its meaning—and continue to revel in its multiplicity through the interactional achievement of people, spaces and things—the ambivalent nature of home-making can be examined as individual as well as collective through projects of identity. I would add that this agential relation is brought to the table as struggle for coherence, and continuity in the work goes on no matter what.

To be clear, I also maintain and reiterate that as a new materialist, cultural space is not defined as around and between objects. It is considered already embedded in these objects (the human–non-human), spaces and things, as well as in spacetimemattering[11]; it is diffractively working and conceptualizing difference through a spectrum of sound phenomena where home is a mutual relation of things and bodies inclusive of diverse participatory powers. As Barad insists, "the acting human subject and the known object are not separate, but entangled."[12] From an ethico-political perspective (with whom and for whom), this relational process (re)models our interdependence with human and nonhuman others, beginning with thinking and acting 'from the middle out' without discrediting the consideration for collectivity and the other.

What emerges from this conceptual relation of the artwork is making the materiality of women's work visible as a performative mode of domestic bodily production: the implicit body of a woman at work is expressed in a resonant sonic overture of resistance. A Public Kitchen is an immersive sculptural installation that is at once a multisensory, embodied, participatory encounter that engenders a mixture of feelings and visual impressions for the participants, such as fascination, surprise, boredom, disgust, perplexity, irritation, joy, and ambiguity. I argue that a Public Kitchen produces noisy affects that can contribute to transforming our social imaginations, forming daring narratives (or sonic recipes) that feed post-human ethical practices and inquiry.

## 4. Co-Composing the Sonic Intra-Face of a Noisy Kitchen through the Apparatus

It is a profoundly intertwined collective moment as both human and non-human elements intra-act within the apparatus. The apparatus can be seen as the framework for the social artwork: a sculptural set-up consisting of the motherboard—a long metallic platform resembling a community kitchen table used for 'placemats' of positioned kitchen tools, electric kitchen appliances and electronic music hardware. Kitchen tools and appliances are amplified and activated through the simple technology of a contact microphone, connected to the metal motherboard. A contact microphone, also known as a pickup or a piezo, is a form of microphone that senses audio vibrations through contact with solid objects and musical hardware. Thus, the transduction produced when kitchen tools interact with this active metal surface generates noise, or diffractive interference patterns, through a piezo electric effect that measures sonic vibration, pressure, acceleration, and the force which converts sound to an electrical signal. As Barad explains, "apparatuses are not mere observing instruments but boundary-drawing practices—specific material (re)configurings of the world which come to matter" (Barad 2007, p. 140). Thus, I use the apparatus as a central concept for what I argue is a new materialist approach, by working with a cluster of participants who collaborate through sound improvisation in performative intra-actions/s via the motherboard to channels of audio-electronic processing. I interpret the apparatus as a middling point for how sound performance reflects the current post-human condition and further feminist concerns. A Public Kitchen does not necessarily look high-tech when performing with kitchen tools and appliances relational to the everyday through the 'doing and cooking'; it is more about creating a condition in the installation that is a Public Kitchen—to generate a state of being for the non-human and performs in a particular temporality in relation to spacetimemattering.

---

[11] In this research, "timespacemattering" refers to how intra-actions emerge between space/time/matter relations and forces within the entanglement of differential relations (Barad 2003).

[12] (Barad 2007).

## 5. (Re)configuring our Relationship with the World in Co-Becoming Processes

Relationally, feminist power relations are located within the live enactment. In contributing to social change, I argue that activating a feminist ethico-politic must fundamentally emerge from work done at the transformative, energetic layer of the body, activated by doing-making processes. Cooking. Playing. Making art. Resting. Working. Sharing touch. Sharing food. Gathering to remember. Growing up. Growing old. Grieving loss and oppression. Resisting. Empowering humans. Performing ritual after ritual as an affirmative politic in action. All of these processes (and more) could be seen to fall into the 'doing-making' constellation.

(Re)configuring our relationship with the world is not a direct, elementary trajectory; it can entail messy and disquieting encounters, which embrace the indeterminate and manifest through negotiation and contradicting emotions. By engaging attentively with a sonic performance installation, a Public Kitchen entangles performers to participate in co-becoming processes, defined by Rosi Braidotti as "a co-operation of active laboring, becoming-ethical" (Braidotti and Hlavajova 2018). The challenge is to change our assumptions about the social structures that shape the world around us and take a deeply critical look at how one actually engages with the very systems that we are trying to release ourselves from, disengage, disentangle. I argue for the need to be brave and name, identify, call out, bring up, and sound out the social patterns that surround us through a sonic recipe, shifting the social dynamics we are entangled in. I argue that these social patterns of sonic textures, both tonal and atonal, maintain a quality or character of *becoming* that can be observed as other-worldly and decorous, by cultivating the experience, or can even result in disillusionment in the active process. "Becoming" here refers to feminist theoretician in new materialism, Rosi Braidotti's idea of it as a rhizomatic political action (as in, rhizomatic learning), proposing that it is a significant undertaking of nomadic feminist ethics in which transformation is to consider a body's potential for mapping and shaping transformation in the local. In turn, Braidotti considers that new futures need to be imagined first and re-assembled elsewhere in order to do *the work*, rather than being constructed into the same hierarchical frameworks.[13]

A feminist politic expands on Fluxus precedents through the use of props, costumes and makeup to create different personae with participants in a Public Kitchen: It is commonplace for performers to be adorned in crocheted masks, A-line dresses made from reflective industrial fabrics, and sticky neon duct-tape overlays. A Fluxus attitude can mean that everything that is material matter, in mind and body, is embraced—from absurd to commonplace, from violent to tedious—to ignite the kitchen space as a site that shifts and repositions its domestic status to a site of materialist agency. For example, the performance art collective, W.W.K.A. (Women with Kitchen Appliances), which existed in Montreal, Canada, from 1991 to 2013, modelled a feminist performance art approach that adhered to aesthetic concerns associated with beauty, free play and feminist autonomy. With the emergence of a Public Kitchen as a speculative and somatic proposition, and creation of a sound performance collective named Sonic Electric (currently positioned in Melbourne, Australia), a more political cultural affective reading of today's feminist discourse, conditioned by our current socio-political climate, is active through an ethico-political practice.

## 6. The Sonic Intra-Face of a Noisy Kitchen Workshop

How can collaborative sonic art work create a Public Kitchen (as in Figure 1)? On this collective journey, sonic material phenomena are created with kitchen tools and electric appliances, co-composed by volunteer participants who are gathered around an immersive kitchen table installation, *the motherboard*. As previously described, the motherboard is an ambulatory surface platform, resembling a lengthy metal kitchen table, that is used for the placement of kitchen tools, electric kitchen appliances and electronic music hardware. It is a vital and central component of a Public Kitchen

---

[13] (Braidotti 2002).

sculptural installation. The sonic performance installation is built on a critical arrangement of hybrid personae (costumed participants) whose bodily capacities are measured by the continuous mutual transformations between 'humans at work,' seeking to reposition the kitchen tool or appliance by exploring its displacement and functionality through the labored body (Figure 2).

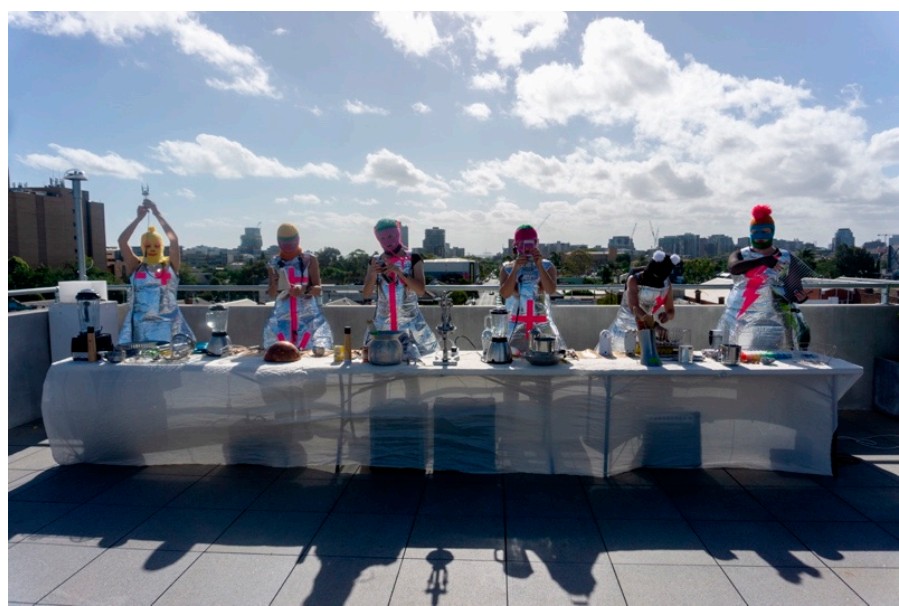

**Figure 1.** A Public Kitchen performed by Sonic Electric at MARS Gallery, Melbourne, Australia, 2019.

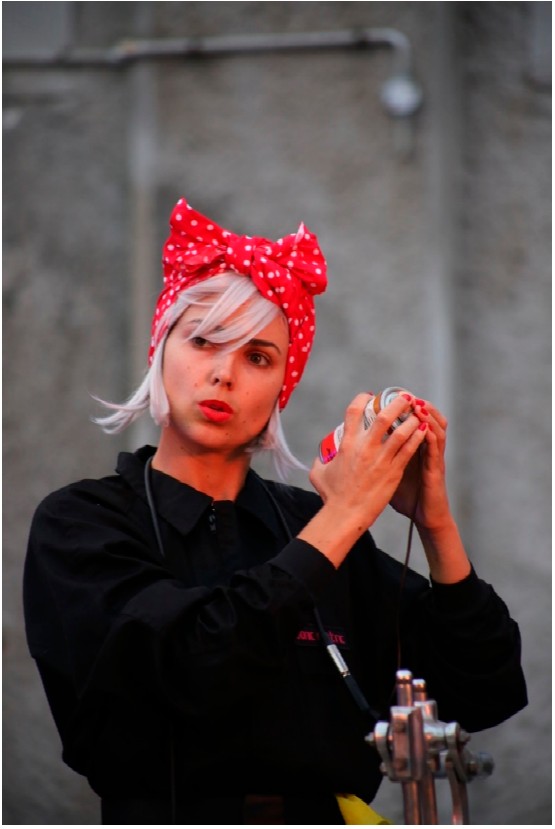

**Figure 2.** A participant of a Public Kitchen performing in *The Future is Female* exhibition, Abbotsford Convent, Melbourne, Australia, 2018.

I argue that investigating sonic materiality through the prism of a Public Kitchen enables the power of sound performance to be activated as a savoring of self-emancipation for the participant. Sonic phenomena re-sounds from one kitchen tool to another—from Australia, Spain, Iceland and Hong Kong—through space, agency, resonance and temporality in the apparatus.[14] All of these critical attributes reverberate in Public Kitchens as socio-cultural phenomena that create points of resistance (Figure 3).

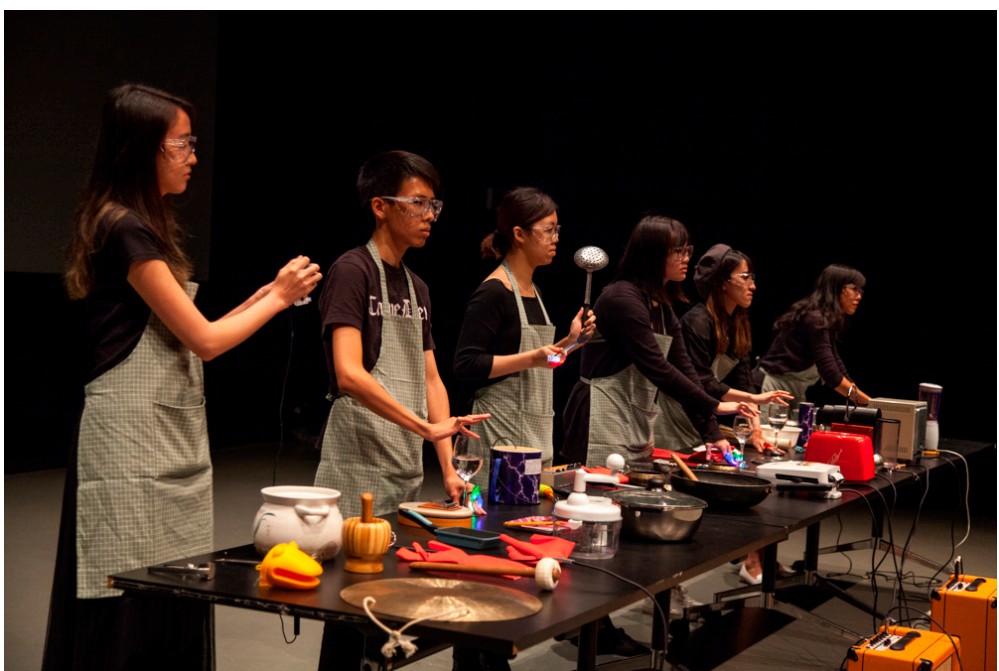

**Figure 3.** Participants of the 22nd International Symposium on Electronic Art. *ISEA2016 Hong Kong* 香港 *Cultural R>evolution*. Public Kitchen. School of Creative Media, City University of Hong Kong, China, 2016.

This awareness is found through a Public Kitchen, with the desire to affect what we seek to change in the entanglement, drawing people of all shapes, sizes, ages, and abilities to understand and measure difference through the feel, flow or vitality of the enactment of sonic intra-actions. This immanent process produces intra-actions and/or fluxes and vibrations of matter, resonating with human and non-human bodies. I argue for a performative feminist materialist ontology/epistemology, which sees social phenomena as primary and non-binary; here, subjects and objects are dependent on forming in or out of a process of intra-action by rethinking the dualisms so central to post-human thinking. The oppositions between nature and culture, matter and mind, the human and the non-human, are produced and combined in the action itself, reflecting on the philosophical writings of Karen Barad (2007) and Rosi Braidotti (2013).

This perspective opens up a capability for deep listening by the participants that makes them available and receptive to a tuning of the world we live in. Resonance can be seen as proposing a new way of thinking about causality and agency that acknowledges the indeterminate possibilities for worldly (re)configurings—a (re)-tuning of the world. Resonance, I would also argue, should be understood in terms of relation: a vibration that emerges out of intra-activity. This raises an awareness of the complexities involved with diffractive paradoxes revealed in sonic relations.

---

14  The public iterations in specific localities in and around: Melbourne (Australia), Blanca (Spain), Reykjavik (Iceland), and Hong Kong (China) are all part of PhD field research work conducted from 2015 to 2019 and supported by The Victorian College of the Arts, University of Melbourne, Australia.

There are diffractive readings in making sound—or, sounding and hearing—where sound/noise resonates in composition through communication between participants: noisiness, the Public Kitchen activity, pronounces a form of resistance or critical engagement with the world. This relation also punctuates the entanglement between noise and a politics of consent in the way a diverse range of femme, female-identified and gender non-binary persons collaborate. A Public Kitchen addresses all cybernetic bodies as forms of noise and disruption that are engaged in a thinking about freedom where the mind/body is a driving force within a multiplicity of becomings; this is emphasized and expanded on through the writings of Braidotti, Gilles Deleuze and Felix Guattari (Braidotti 2013; Deleuze et al. 1987).

*Understanding the Transducer*

Sound patterns as sonic recipes in a Public Kitchen are formed by human participation with the help of a small electronic audio device—the contact microphone (Figure 4). Contact microphones create live contact with kitchen tools and appliances entangled within the apparatus through the motherboard. Lead cables connected to contact microphones feed guitar amplifiers placed under the motherboard, amplifying noisy material into a public space, and leading the listener on a sonic journey. It is a haptic partnership with musical hardware and bodily production that, in concert, produces sonic performative phenomena. I argue, as a multi-media visual artist, that the strength of sonic phenomena is that it does not strive to be technically perfect, musically high-tech, or audio-efficient. Instead, it accepts its place as indeterminate, vital, resonant material, celebrating its flexibility to jump from manual labored sound to electronic sonic space, and vice versa, in a sonic recipe. There are constant exchanges and transformations happening in every kitchen: forces shifting; particles entangling with particles in the quantum field of thinking; doing and making in the world in paraphrasing Barad to explain things on a molecular level (Barad 2007). These intra-actions are technologically entangled with musical hardware, constructed with industrial metal parts. Music hardware is mechanical objects or electronic devices that create, or aid in the creation of, experimental musical sounds that a musician might use to enhance *sonic material phenomena* in a live performance.

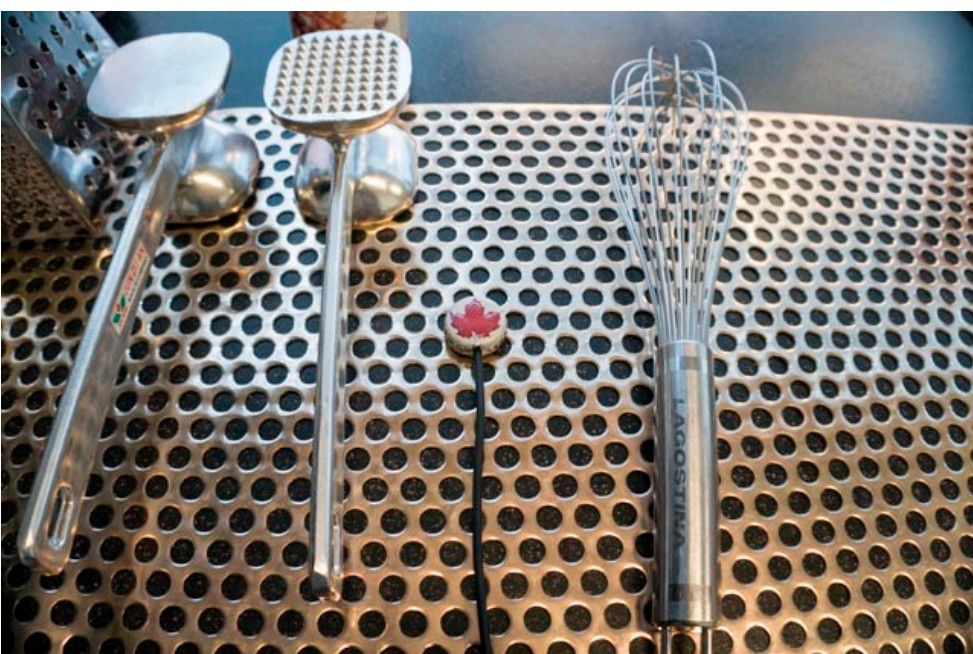

**Figure 4.** A contact microphone placed on the motherboard surrounded by kitchen tools. Testing Grounds, Southbank Arts Centre, Melbourne, Australia. From the sound performance work, *Being in Time: Death by Audio*, 2018.

A Public Kitchen dares to take the risks proposed by Braidotti in this paper's opening quote, as it reflects on the uncertainty of our current times to expose social rifts and find affirmative human–non-human connections in co-composing sound patterns into sonic recipes. The material realities of women's everyday lives are reflected in a Public Kitchen as, historically, we can see the kitchen as a place that separated people, as a place of oppression, a key site where women perform most of the world's labor (International Labour Organization 2016).

## 7. All Art Is Political

Materialities are never neutral. Thus, a Public Kitchen could be conceived as an alternative economy critical of a capitalist system. I argue that all art is political because a critical feminist perspective articulates that patriarchal culture is a system that impacts everyone as material conditions of all sorts play a vital role in sonic resistance against domination. Domination in today's cultural politics takes various forms in varying social contexts.[15]

As explained by Rosemary Hennessy, materialist feminism considers how women and men of various races and ethnicities are kept in their lower economic status due to a power imbalance that privileges those who already have privilege, thereby protecting the status quo (Hennessy and Ingraham 1997). A Public Kitchen challenges this propositionally, as it manifests a (re)building of community in the form of collective action that is unyielding through sonic material resistance, playing out as a collective pathway of negotiation and co-operation (Figure 5). This position excludes a Marxist agenda in a Public Kitchen by recognizing the agency of biology or matter in worldly phenomena and social and political human behavior. Feminist new materialists challenge the linear models of causation that form constructivist analyses of the ways power relations shape the subjects and objects of knowledge. This occupation accounts for how intra-actions through which the social, the biological, and the physical emerge, persist, and transform. Feminist new materialists are qualitatively trying to change feminist critical analysis from a framework within which the agency of bodies and material objects is understood as an effect of power to a framework within which relationally, nature-culture and biology have reciprocal agentive effects upon one another. In this material turn of the distinctive and effective agency of organisms, ecosystems, the human non-human as vital matter, feminists will rethink how to take on social justice, creating paths toward social and political transformation.

Relationally, the monotone video work, *Las Atrevidas: The Risk Takers* (2015), monitors the mechanical movements of six senior women from a small village in Blanca, Murcia, Spain, who embrace tradition, a nomadic culture through difference, and, via their participation, generate a space of subjective individuation and otherness (Figure 6). I investigate how kitchen tools are passed on socially and somatically, stretching entanglements with shifting measurements of performative movement, sonic undulations, historical kitchen knowledge, and female genealogies in the domestic sphere. I argue that, for feminists like myself, temporality exists by how time is being kept, by whom and for whom in a Public Kitchen (van der Tuin 2015). The entanglement of "doing-cooking" transfigures, as Braidotti specifies, how objects resonate as matter that thinks and feels as an extension of the human body through the post-human (Braidotti 2013). The relationship between subject and object opens complex nets of intra-connectivity in which the body, space and psyche are never conclusive and always "more than" something: as Erin Manning explains, they are "indeterminate" (Manning 2016, p. 14). A Public Kitchen presents how this awareness of kitchen objects, disciplines

---

[15] This idea points to a "Trumpian" autocrat world, such as heteropatriarchal domination, that is, a socio-political system in which the male gender and heterosexuality have primacy over other genders and over other sexual orientations, neoliberal domination, racist domination, and homophobic domination, etc. Donald John Trump is the 45th and current president of the United States of America, who took office in January 2017. "Trumpian" is a slang definition found here: https://www.dictionary.com/e/slang/trumpian/ (LLC 2019).

and practice are always operating in relation to matter that is waiting to be interpreted, rehearsed, repositioned, and transmitted (Barad 2003).

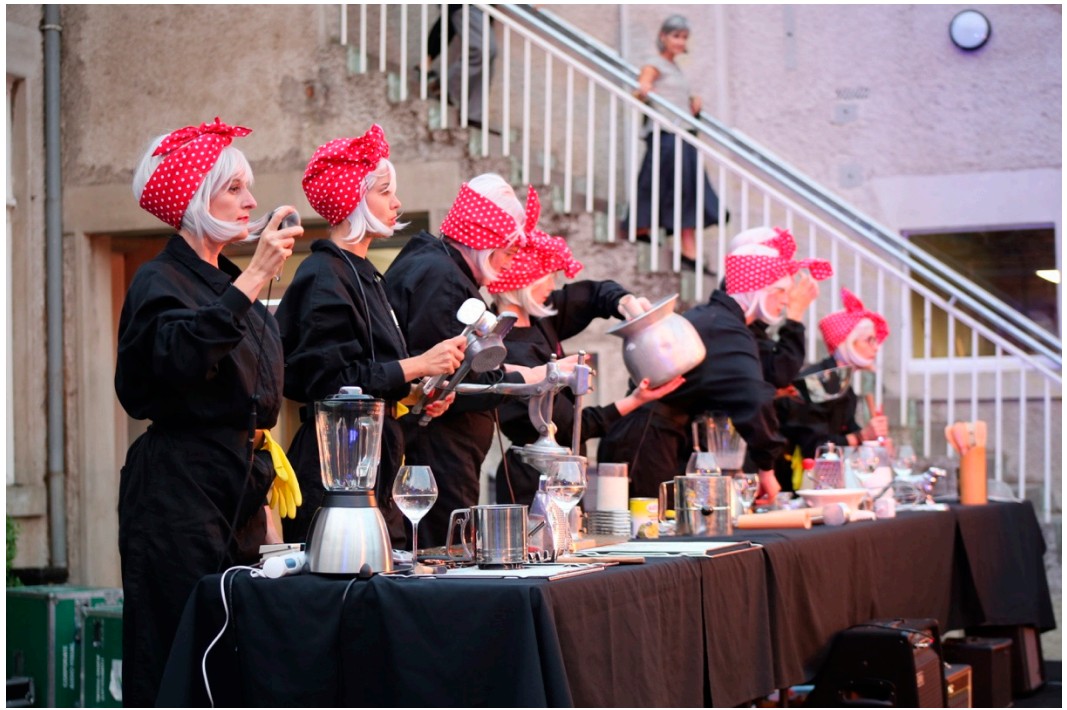

**Figure 5.** The local sound performance collective, Sonic Electric, performing in *The Future is Female*, Abbotsford Convent, Melbourne, Australia, 2018.

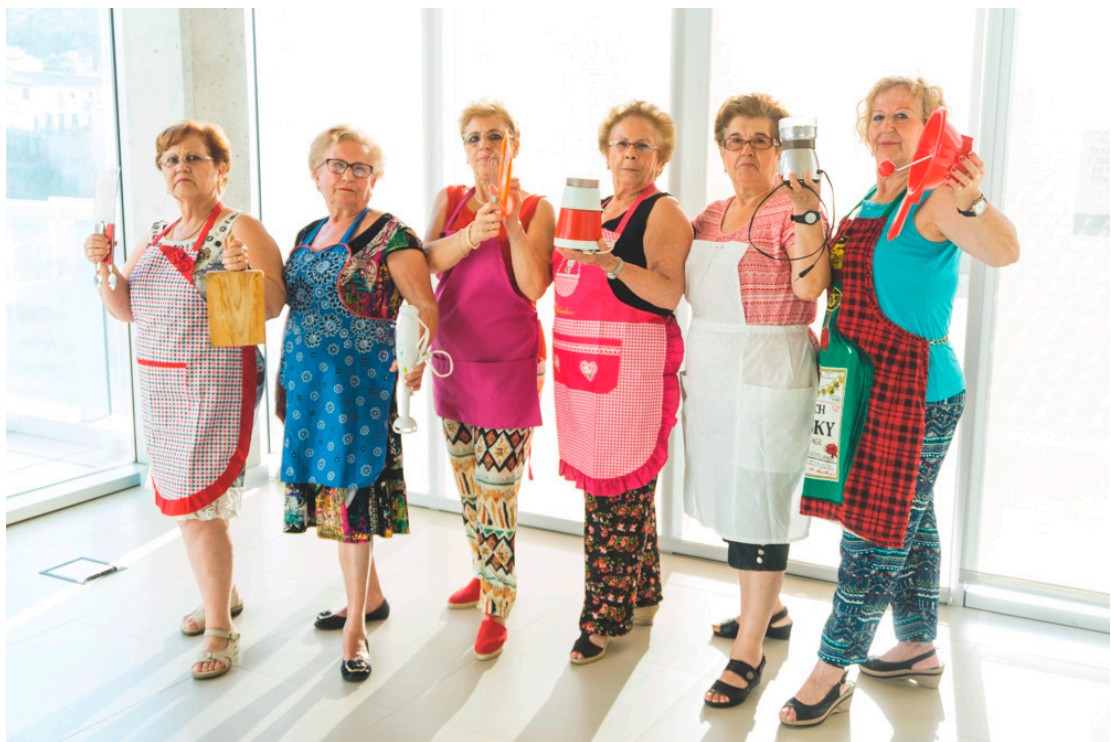

**Figure 6.** *Las Atrevidas: The Risk Takers*, Artist Residency at Centro Negra AADK, Blanca, Murcia, Spain, 2015. Group portrait. Video: duration 10 min.

The practice of deep listening gives women an opportunity to explore their/our bodies and their/our temporalities from women's experience as women while also being inclusive of gender-fluidity. Deep listening is also termed somatic listening by Pauline Oliveros, who observed that the practice of deep listening is a commitment to cultivating receptivity through the body with an emphasis on inclusiveness of performing music.[16] Everyone (women, men, non-binary, trans, gender fluid etc.) is engaged as participants in this practice before even touching a contact microphone. For the participant, inclusion promotes a veritable, personal movement of a thought and listening practice that helps the body unwind while at the same time stimulating awareness of energy flows of the body, quite similar to a Tai-Chi or Qigong class that incorporates Chinese medicine and the flow of chi (energy) points throughout the body. In the company of others, this approach prepares a participant foremost to listening in synchronicity with each other in performance, relational to the whole apparatus of doing-cooking as an unfolding immanent process.

Feminism today is a worldview that includes men, demanding a rethinking of power in society, a change in the dominant system of private domestic space, public space and temporality; negotiating a woman's place in art institutions and valuing a woman's place in this discourse (COST: European Cooperation in Science and Technology 2017).

What feminist new materialists point out is, what is in the world and what we know about things in the world cannot be considered as different things. What is in the world, what we know about things in the world are constantly shaping one another and, in the kitchen, this study of matter and meaning is, moreover, boiling over. Thus, the tools of the kitchen become agential matter in a post-human world. Post-humanists are responding to the redefinition of humanity's place in the world by both the technological and continuum in which the "human" is but one life form among many (Braidotti 2016).

## 8. Semiotics of the Kitchen

The socio-political oeuvre of American artist and activist, Martha Rosler (Rosler 1975), highlights her position on the artist's role within contemporary art practice, as well as her understanding of "private to political." This position affects and mentors the present understanding of the role of an artist as a feminist thinker, and this agential strategy energizes the creative move from the private, to political, to public space present in a Public Kitchen.

Rosler's seminal video and performance artwork, *Semiotics of the Kitchen* (1975), contributed to a recurring thematic during the 1970s that still serves as a critical document and historical precursor to current feminist materialist discourse and genealogies. In this work, the artist critiques women's roles in society by acting as a 'cooking show host' personality, creating an alphabet using different kitchen tools. The letters are articulated, often quite violently, in the process. Through Rosler's objective lens, the kitchen table and its environment are made visible as a platform of domestic labor: a form of maintenance work, household chore and food preparation. It is coded into a form of semiotics—a performative and gestural language—acted out through the aggressive, violent and monotone handling and manipulation of kitchen tools.

Rosler developed her own terms of performative engagement in front of the recording camera to articulate a feminist debate. And while this seminal work seemed to focus on her personal place in the world, Rosler's critique spilt beyond her 'artist's kitchen' to examine the political context and capitalistic economy of the late 1970s in the United States. However, her analysis didn't end there. Rosler was also accounting for the personal, bodily and psychological experience of being human. At the time, her semiotic approach pointed to a Marxist–feminist expression of frustration as a radical mode of feminist critique in that it stirred the global soup of women's oppression. Rosler's message was not only directed to the women (the Other) through which feminism defines itself, but also to the forces of government in an advanced industrial society (Molesworth 2000).

---

16    (Oliveros 1973).

*Semiotics of the Kitchen* politicized the space of the kitchen through the lens of the camera, exposing issues of consumer culture, mechanized labor and material handling as a monotone assembly line. Through performance art, Rosler was steadfast in taking a stand on government in which political authority exercises absolute and centralized control through consumer capitalism. Her work has influenced the embodiment of physical labor in the kitchen and speaks especially to a Public Kitchen model, highlighting how politics and the performative can be embedded in the very material presence of video art and an alternative politics of connection. At its very center is the materiality of artistic and political action. It shows how artists can navigate this territory, giving attention to the question of politics from the perspective of theory and the artwork itself (Nicholson and Seidman 1995). This noisy platform resonates to enable the dictum that "all art is political." In the last decade, there has been a series of incremental shifts to artworks that primarily utilize forms of collaboration and participation on the ground and as live art forms. From a feminist new materialist view, physical labor is highlighted to create a form of resistance or post-human critical engagement with the world; however, the actual power of kitchen tools and appliances to create sound material to connect and contribute to the feminist discursive can build on Rosler's original ideas.

Critical reaction to collaborative practices of social engagement is always problematic in the way it is contextualized by Rosler's socio-political stand, for example. Rosler has consistently made me think of the potential to be had in developing an ethics of engagement in relation to feminist collaborative art practices. Sonic artwork can materialize in form and in practice, with such simple tools connected to electronic media and to the familiar environment around us. As a method of social engagement, a Public Kitchen seeks to create a form of creative occupation, moving from the kitchen into a public space, to create other types of kitchen environments or iterations with people other than myself. The term *iteration* is used in a single execution, as of a set of instructions that are to be repeated in a performance work. As a diffractive reading, performative gesture and the semiotic language have been extended beyond Rosler's seminal work. The kitchen table shifts in time, space and location. Each kitchen tool and appliance that is placed on the kitchen table surfaces from a set of distinct kitchen tools and skills. Each tool is picked up and transformed in live performance to become a sound instrument in its engagement with electronic media, not only with manual kitchen tools, but also with electrical modern kitchen appliances.[17] The physical body is bound up in this process (Braidotti 2002).

Therefore, a Public Kitchen shifts and repositions kitchen tools and appliances in ways that seek to open up sites of resonance and resistance. Typically, noise from the kitchen is considered unwanted—it is something extra and excessive. What is considered noise or information in human terms is seemingly processed by digital technology in an equally blank way; a problematic appearance that can hide human accountability behind the apparent autonomy of technology. This paper considers this posturing through the way communication is made noisy, and by addressing all cybernetic bodies as forms of noise and disruption that are engaged in thinking about freedom with the mind/body as a driving force with a multiplicity of becomings (Braidotti 2002, 2013).

Braidotti calls this interaction a transmutation—a qualitative change, a metamorphoses—"towards a materialist theory of becoming" by using the power of technology in an affirmative way (Braidotti 2002, p. 1; 2013, p. 21). And, like Rosler's strategy, physical labor remains an extension of the self (the participant), except now there is more than one person occupying the space at the kitchen table as collective affirmative action becomes a platform for collective, cross-disciplinary inquiry for both art and social commitment.

---

[17] It is important to pinpoint the probability of coming across unusual kitchen tools that are kitchen staples in the country from which the iteration takes place, and to distinguish the variety of different kitchen tools that are used in a Public Kitchen across the motherboard. Each kitchen tool is best used for certain foods with respect to ethnicity and locality, and the importance of using specific utensils is a guide to the gestural technique that is used by the participant.

*Bodily Encounter*

In Rosler's *Semiotics of the Kitchen*, the performer weaves a personal, psychic and physical encounter into the narrative. This individual encounter is critical in the handling and manipulation of kitchen tools and appliances used in food preparation, domestic labor and maintenance work to manifest a political public artwork. Rosler's artwork was explicit in her black and white, severe characterization, set in her kitchen in Brooklyn, New York. Rosler posed in front of the camera as a homemaker, picking up a kitchen tool for each letter of the alphabet and naming each tool with a violent gesture while staring, deadpan, into the video camera. At the time, the creative phenomenon of Julia Child[18] was appearing on local public television channel, PBS. Rosler, starring in her own reality cooking show in the space of her own kitchen, demonstrates the object's potential, not its culinary purpose (Figures 7 and 8).

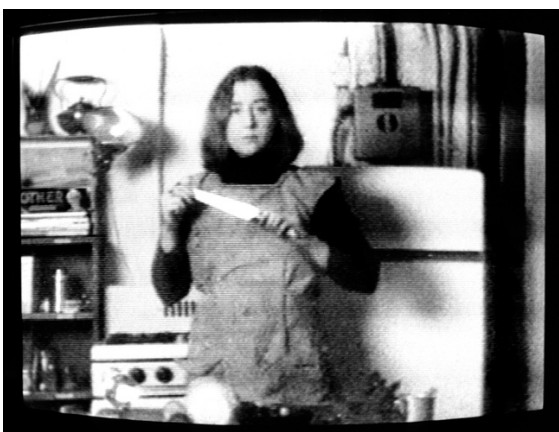

**Figure 7.** Martha Rosler, *Semiotics of the Kitchen*, 1975, video, Video Data Bank, USA.

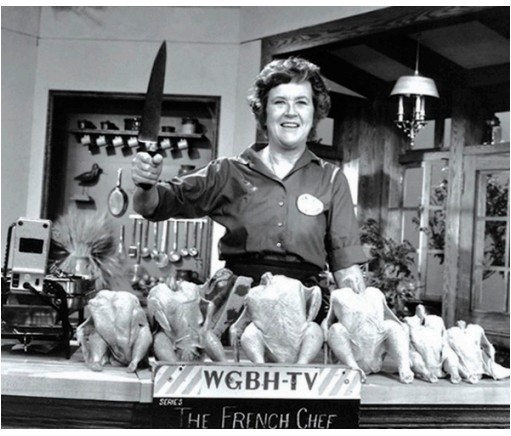

**Figure 8.** Julia Child in *The French Chef* on WGBH, 1963–66. Photo: Paul Child/PBS television, New York, USA.

As Julia Child so often presented through her TV personality, a kitchen tool was usually handled with skillful culinary expertise; but Rosler's deliberate physical gestures were adamant, violent and rebellious. Rosler stabbed at the air with a knife and fork as she cast off ingredients behind her into the void of the kitchen and beyond. Rosler returned her gaze to the camera lens as a mad housewife, not quite the elegant French chef.

---

[18]　Julia Child revolutionized American cuisine through her PBS television cooking shows, such as *The French Chef*, *Cooking with Master Chef*, etc. Her videos are available online.

Rosler's subversive projection of a woman's place in the world signified a desire for resistance and change. However, it also signaled failure, as the artist portrayed herself as a prisoner of her own domesticity by mocking Julia Child's persona. Fast forward. In a Public Kitchen, the performer also adopts a persona. However, foremost, the aim is to underscore the role of the participant as a sound maker, a collaborator, to create sound patterns that can resonate as a social and political marker for resistance and change but can also signal the failure not to do so. The sound material that is collected will be variable. Its indeterminate pathway will either fail or succeed in its outcome and potentiality.

*Semiotics of the Kitchen* is testament to today's continued addiction to entertainment in the form of cooking shows and the world of celebrity chefs. Television media has amplified the artifice by which social media and the power of television homogenizes and distracts from meaningful social action. Rosler employed parody in the way that cooking shows inhabit our screen and social consciousness: she deconstructed gender, tracing how feminist genealogy pays particular attention to discourses, bodies and power. In a Public Kitchen dynamic, the participant is encouraged to engage in over-the-top theatricality, or to mask psychological or emotional identification. In this way, they play out, as Rosler did, with deadpan humor and passive aggression in order to reflect society's pull of conformity and homogenization.

## 9. Resistance through Noisy Resonance

> "Drama is very important in life: You have to come on with a bang. You never want to go out with a whimper." (Julia Child to Jacques Pepin during *Cooking in Concert* television program (Child and Pepin 1996).

Noisy resonance emerges at a vibration of larger amplitude when produced by all participants playing a kitchen appliance in a sonic recipe. It is this noisy crescendo that fills the space of a Public Kitchen. Materially, it becomes a resonating system of vibrational bodies inclusive of a kitchen object's interference patterns. Resonance, scientifically and acoustically, occurs when a system preferentially vibrates at a certain frequency. This frequency is called resonant frequency, and the system will respond very strongly to any periodic force at that frequency. The exact frequencies at which objects resonate is largely determined by the object's physical properties: its size, shape and the materials that it's made out of. Many objects have resonant frequencies, and they are the source of many of the sounds we hear. When you knock on something, much of the sound you hear is just the ringing of that object's resonance. Even our own bodies have many (Pyzdek 2018).

Sound resonates through human sensory behavior experience by sculpting, shifting and changing the perception in which the body labors to listen creatively to objects and other bodies vibrating; this process is critical of power relations in the kitchen environment. A field of resonance is important to me in the material discursive and the transversal[19] (Rhoades and Brunner 2010). Resonance strikes a chord in a Public Kitchen in the act of reclaiming thought or sound which is important for feminism, new materialism and my own practice. Sound, relationally, works its way to the forefront of contemporary sensory behavior and user experience by sculpting, shifting and changing our perception of the kitchen environment in which the body labors to listen creatively and critically.

In new materialist thinking, resonance can replace the binaries of structuralist thought, shedding new light on contemporary debates between sound, aurality, cognition, subjectivity, and embodiment. I argue that this is due to resonance's ability to dissolve the binary of the materiality of things (human-non-human), and compels us to call into question that something, such as resonance, must therefore situate itself as a form of resounding together in the discourse of post-humanism and other

---

[19] Transversality addresses different existential territories and universes of value through the register of resonance. (Guattari 2003). "Psychanalyse et transversalité". Paris: Maspero/La Découverte. "Transversal Fields of Experience" brings different points of entry into resonance that all revolve around the question of how we open up new registers that incite a creative moving with the forces we encounter in contemporary transversal fields that shape our everyday experiences (Rhoades and Brunner 2010).

immersive feminist participatory practices. For many feminists and artists alike, labor relations and ecological issues become more crucial to our survival every day. In a Public Kitchen, I see sonic connections of shared affective and productive movement to understand new material physicalities, new emotional transformations and new sonic relations, intra-culturally, with others and the world. A Public Kitchen takes affirmative action into the streets as a form of resonance. The kitchen object is relational, in that is has the capacity to make felt how an object is already in a field of relation and tuning through intra-action, intensities and symbolic forces of resonance. A feminist new materialist position illuminates a cartography and radical visibility for challenging theoretical concepts of art and everyday life, work and value for individuals, women, communities, and women's oppression in the global environment.

Therefore, it is important to discuss or consider the ways in which different discourses, such as affect, mingle with matter and virtuality in the kitchen. Through this discussion of the artwork of Rosler and her social-political connection to the materialization of the labored body, as well as my critique of a Public Kitchen, I aim to find a modulation, or flow, of affect in the discursive that never jumps clear of its entanglement in the processes of new materialist performativity. Instead, it assists in unpacking and defining resonance and power relations, enhancing the position of feminist genealogies.

Working on transnational participatory collaboration is a real process demanding particular concepts and commitments. Teaching sound through deep listening is a practice. Creating a visual aesthetic of performativity within the immersive assemblage is also significant. The entanglement of bodies and the sound that is transmitted, are foregrounded in a resonant system. Through this process and apparatus, it becomes clear that what is transcended through sound is the understanding that the enactment is re-arranged with every iteration and is always evolving.

Contact microphones and electronic media move in the mind, through the body and through things as resonance is heightened, and, as Manning characterizes, develop into something "more than" movement: "It is out of time, untimely, rhythmically inventing its own pulse" (Manning 2016, p. 2). Thus, a sonic performance work makes public the sound material that is emitted from kitchen objects—a new materialist way of listening and behaving, not *in* but *of* the resonant world.

*From Failure to Empowerment*

American artist, Mierle Laderman Ukeles, writes in her exhibition proposal, "Manifesto for Maintenance Art, 1969! Proposal for an Exhibition CARE", that "Maintenance is a drag," confirming that maintenance is unglamorous, tedious, hard work and economically underestimated (Ukeles 1969). In a not-so-perfect world, something is always breaking, wearing down, getting dirty or falling apart. In the performance, this lopsidedness and inequality is transmitted as an embedded and entangled process of coming into, working through, being formed by, and forming something into a sound recipe—a set of sound patterns created by participants. The unevenness of this process in sound and noise art is relevant to how the participants, at times, are almost failing at getting the object to make a sound or trying to change how the object can be rendered mute/useless in its repositioning. The sound, as a physical phenomenon, is transduced into an excited state, becoming dynamic as it is electrified, as contact is made with an electronic contact microphone. When the contact microphone is removed or disconnected, the object returns to its static self again. This is a complex relationship where human, non-human and so many natural, social, political, and cultural factors are forces that resonate in the entangled processes of materialization. Surely 'failure' is recognized as agential forces that activate an acknowledgement of nature, the body and materiality in matters of uncertainty—indeterminacy within the intra-action? This failure slippage plays simultaneously into the timeline, where indeterminacy of position and momentum in the sonic compositional arrangement manifests as a communication breakdown. It appears, within each diffractive reading, as a disconnect, or simply the object responding in an auditory relation as if it is being agitated in the failure to produce a resonance, or keep going, or fade away.

I would shape this disruptive mode of engagement in consciousness and thought as creating resonant agitation with kitchen objects in the radical potential of the political. This agitation is defined by a discomposure or disturbance that is transmitted through resonance, such as through the delivery of a feminist manifesto. To me, agitating space with sound and noise art demands a new way of understanding participatory practice, audience perception and difference in manners of expression and modes of experience of the artwork.

Akin to Rosler, I lived in New York City during the 1970s and was exposed to performance art, punk and hard-core music, which were important influences for many local artists. Girl power, or GRRRL power, acknowledged a woman's space in the world of music and the performing arts.[20] The Riot Grrrl subculture emerged in the early 1990s from the punk rock scene during the third-wave feminist movement, uniting women and girls against capitalist and patriarchal cultural ideologies. Countering the dominant ideological narrative in the United States, the Riot Grrrl movement continues to evolve and expand to avoid commodification. Moving between high and low art, the relevance of *sound as social* with feminist and political concerns and polarities is activated in a Public Kitchen through a bricolage of incongruent cultural elements that become ripe for exploitation.[21] This attitude was rampant in New York City from as far back as the birth of Happenings on the Lower East Side in the 1980s, and the subsequent significant and pugnacious punk and hardcore scenes. Rosler may have taken note of this local scene, as many of these bands had humorous elements to them. The performers dressed in ludicrous outfits, engaged in slapstick routines and experimented with different styles of experimental noise-making. The desire to create music with different tools demonstrates John Cage's celebration of noise and accident ([Cage 1961](#)); it is also informative to note that he was teaching at the New School in New York during this time. This methodology refers to putting sounds together in a random, indeterminate manner; mixing and matching unrelated sounds, noises and breakages in performance.

It was this very combination of moving between musical styles and art practices that led to a productive cross-fertilization between music and art, not only in New York and London, but subsequently in Berlin in the 1980s and beyond. By using a non-musical interface, new expressions were generated. A Fluxus attitude of the 1960s reflected performers dressed in these beforementioned ludicrous outfits, engaging in slapstick routines and experimenting with different styles of noise-making to attract attention to what was happening on the streets and in social life ([O'Dell 1997](#), p. 41) (Figure 9).

*Lunacy* is an interesting word in contemporary ethico-political discourse. It takes special skills to perform and participate in performative politics today in a "Trumpian" world of conspiracy theory, gas lighting and fake news being a coercer in enabling alt-right rhetorical propaganda.[22] Lunacy is the new norm, responding to the political climate, serving up more dead-pan humor. For example, a performative pandemonium recently played out at the White House, Capitol building in the USA on 5 September 2018. A much-anticipated hearing of Judge Brett Kavanaugh's Supreme Court confirmation before the Senate Judiciary Committee was stalled by mostly feminist activists: In particular, they were protesting Kavanaugh's nomination due to his conservative stance on abortion. Outside the hearing room, female activists dressed as characters from *The Handmaid's Tale* (2017)—an American dystopian drama web television series created by Bruce Miller, based on the 1985 novel of the same name by Canadian writer, Margaret Atwood ([Miller 2017](#)). Inside, more than 70 protestors were arrested for yelling and disrupting the proceedings. It was probably the most confrontational Supreme Court (SCOTUS) nomination hearing in recent memory. I would arguably call it a performance art act using

---

[20] "GRRRL" is a slogan that encourages and celebrates women's empowerment, independence, and confidence. The slogan's invention is credited to US punk band, Bikini Kill, who published a zine called *Girl Power* in 1991.

[21] The Swiss artist, Christian Marclay, has acknowledged that during the Fluxus years he was also influenced by the ridicule of, and play on, the formal presentation of music in the concert setting and the live act ([Kelly 2009](#), p. 151).

[22] lunacy (as in "folly") n.: foolish or senseless behavior. 2018. *Nisus Thesaurus*. Princeton: Princeton University Library. Gaslighting is a form of psychological manipulation that seeks to sow seeds of doubt in a targeted individual or in members of a targeted group, making them question their own memory, perception, and sanity.

affective political affirmative tactics. I would also posit what Braidotti distinguishes as "a break with the *doxa*": the acquiescent application of established norms and values by de-territorializing them and introducing an alternative ethic flow (Braidotti and Hlavajova 2018, p. 224).

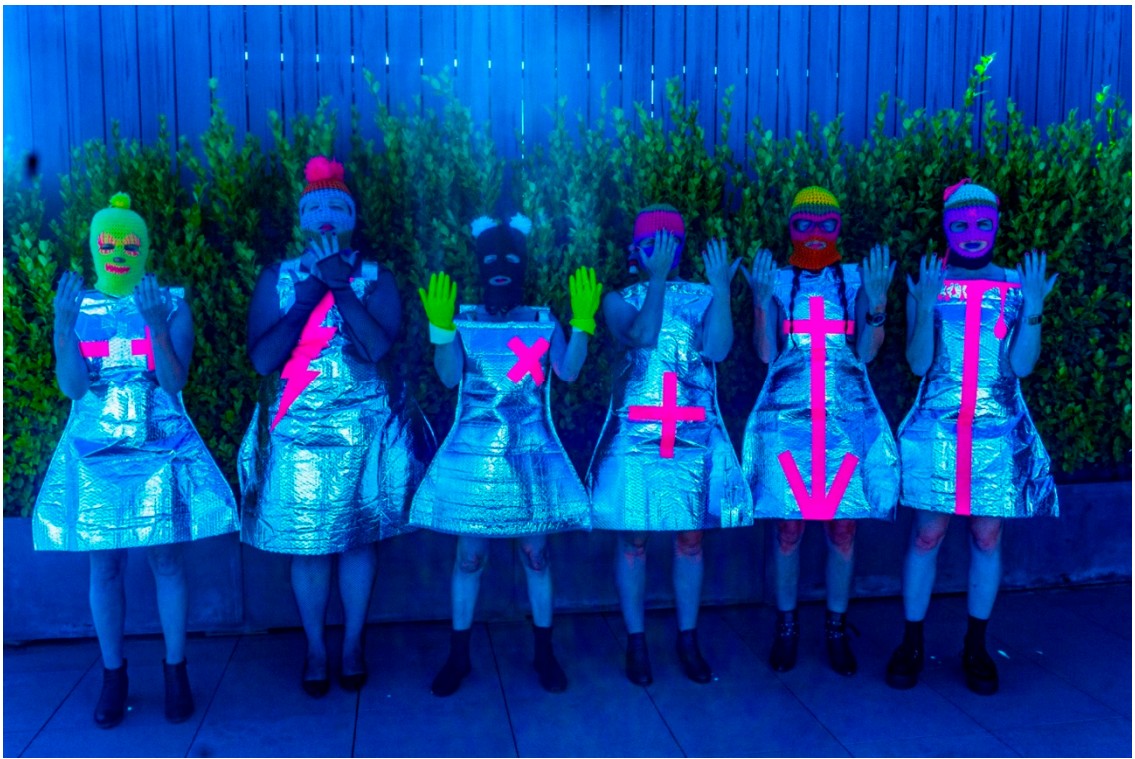

**Figure 9.** Sonic Electric sound performance collective led and choreographed by Juliana España Keller 2019.

There is a dangerous seduction in picking up kitchen tools dressed as hybrid personas. This experience develops an understanding of the many facets of ourselves, thus enabling personal transformation, social dissonance and sonic resonance (Figure 10). There is truth to playing in unity, expanding on *all that matters*. There is political power in the participants' participatory position within a feminist group collective that is played out through the Public Kitchen. Many artist participants would agree to meddling with conceptual standards against which political performative art playfully pushes. As each participant dons a synthetic wig or a crocheted facial mask, the body is the mind in this assumed persona. A dissociation from Cartesian mind-body thinking is also heightened because it breaks down gender barriers as much as it reinforces an idea that you put on a gender, like a change of clothing, immune to utopic thinking. As the punk movement grew out of a drab and dark environment—with participants ready to challenge the status quo and show their contempt for government, society and tradition—it is amusing that the counter-culture term "punk" is now part of mainstream clothing design. Originally intended as a destruction of fashion—both at the literal level through the defacement and damaging of garments, and at the symbolic level via its anarchistic attitude and often blasphemous message—this form of dystopic dressing up enables a sense of freedom and relief to those who participate in a Public Kitchen iteration.

This performative approach can be looked upon as a quantum thinking response[23], not only to what we want in the live performance but who we are being, shattering boundaries between mind and

---

[23]  According to Barad, the deeply connected way that everything is entangled with everything else means that any act of observation makes a "cut" between what is included and excluded from what is being considered (Barad 2007, p. 175).

body. The earliest iterations of performance art, tinged in nihilism (the Dada movement, for example), flirted with the anarchic meaninglessness of language in the early twentieth century (Cohn 2008). The Futurists developed a rowdy theatrical tradition of declamation and noisy musical accompaniment (Cohn 2008). A feminist political rancor is a communal heartache demonstrating the noisy politics of everyone's oppression. The inventory of kitchen tools and appliances of a participant's choice, some from their own kitchen, are placed in front of them. It is as if a cacophonic supper of a specific recipe from a TV cooking show is about to be prepared, but this time it assumes a dystopic fervor, similar to the one witnessed in *The Handmaid's Tale* (2017). It becomes a noisy spectacle of human locomotion, of bodies made into objects extended by a sonic apparatus.

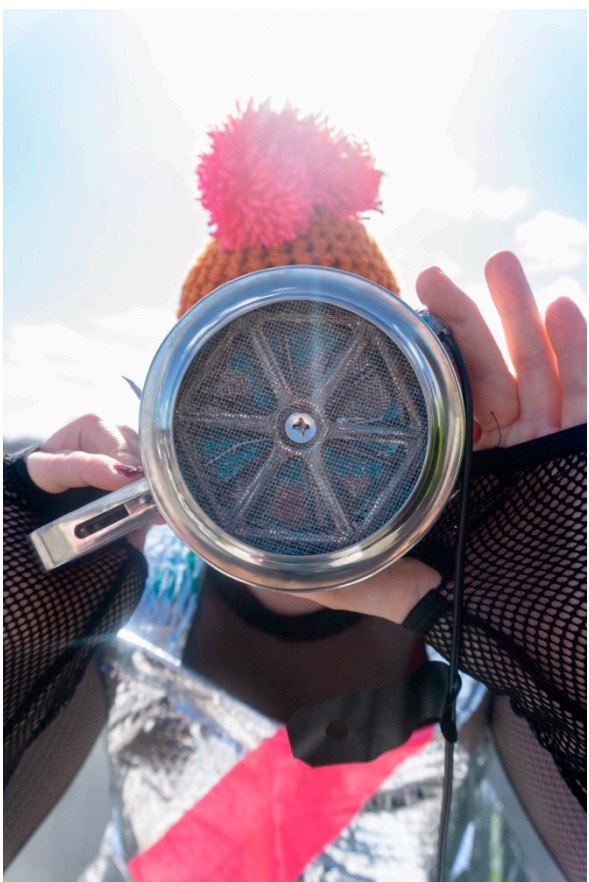

**Figure 10.** A participant of the group, Sonic Electric, re-positioning a flour sifter in a Public Kitchen, 2019.

## 10. Conclusions

A sonic performative live art experience is never fully translatable and has the potential to articulate many perspectives. I conclude that the nature of sound performance art can be articulated as both metaphysical and material. Sound performance needs to be heard to be experienced because there will always be a surplus of sonic meaning and affect which defies containment in any medium other than the sound itself. I maintain that a Public Kitchen is a live interpretation of the social experience of a sound performance artwork. In this interpretation: human machine learning, automation, representation, and uncanny choreographies oscillate between humans, diffractively, with the complexity of mechanical gestures of the somatic body. I use speculative processes, partnered with material phenomena, chemical and physical forces, and intensities, to magnify time-based acts such as sound performance in both research and practice. In this paper, or in a Public Kitchen, I track movement between the human and non-human through intra-actions of the labored body: this is done by emphasizing dramatic transformation as a new materialist politic of connection in a post-human world and future social

robotics as a platform for collective, transdisciplinary inquiry for art and social action and a way of understanding the world from within, as a middling, as much as a part of it.

It is challenging to move nomadically. The entanglement with new materialism, and the engagement in post-humanist thought raises the stakes in the ethico-political consideration of the paths we should consider taking as a species through feminist participatory practices. The artwork, in relation to performative art practices and research-creations offers something more in the arena of public debate and pedagogical practices and makes ordinary participants capable of creating and transforming their own world through a freedom that is distributed individually or by the collective interpretation and practice of a sonic performance. This situated knowledge is critical, creating a rhizomatic relationship between the personal and the collective, validating authentic traditional kitchen tool knowledge around doing-cooking. Somatic learning makes the unconscious conscious, and in the process, leaves one with more options for moving, acting, thinking, and living. It maximizes not just the physical body but one's full human potential.

To reiterate, Braidotti asserts that a philosophy of the body is long overdue in the humanities to address women, gender, critical race, science, media, culture, and animals despite all the clichés we have in mind about femininity or feminism and transdisciplinary practices. I propose that I attend to difference, diffraction, and affect in knowledge production and feminist thought, intensities, emotions and somatic gestures, always being open to the more-than human towards a feminist ethico-politic. I argue the artwork forms a rupture, a collapse in the everyday making of semiotic codes permissible in the performing (the doing-cooking with kitchen tools) in the kitchen in a processual context and works hard for social justice.

A sonic recipe does not fix the post-human world but creates a noisy culture of social (re)imagining. It is a becoming of being, entangled in things and technologies, contributing to feminist genealogies and ethico-political practices. Our bodies depend on food from, in, and through the world of a kitchen environment, and our societies are built on and through things such as kitchen tools and appliances. This is how a Public Kitchen thrives.

**Funding:** This research received no external funding.

**Conflicts of Interest:** The author declares no conflict of interest.

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
