# Peer review of "The Sonic Intra-Face of a Noisy Feminist Social Kitchen"

_socsci, doi:10.3390/socsci8090245_

Round 1

Reviewer 1 Report

Dear Editor/s

Thank you for asking me to review this paper. I apologise for the delay. This paper merits publication. 

The premise for the paper is very interesting, hence my interest in reviewing this paper. The author/s state that the focus for the paper is unpicking a socially engaged art practice, Public Kitchen, through a new materialist lens and the intra-actions of sound and bodies and the agency-pedagogy of sound as matter. However, while the content is highly engaging, there are areas that in my professional judgement need to be addressed.  

Title: grabs the readers attention however, I question the use of interface specifically if you are using new materialist theories to underpin- could this be reworded to intraface?

Abstract: Is overly jargonistic- sentences do not link, and the flow-coherence is  hampered. Need more straight forward communication regarding what, why, how, so what- and ensure sentences link-flow.

Why start the paper with a quote and then not explain-expand-clarify. while this an academic writing genre convention- I find it does not add anything to the paper.

Introduction: The content after the sub-heading introduction is very looooong and is not functioning as an introduction to the premise-content-argument of the entire paper. I suggest the author/s need to write an introduction and include before this content with a new sub-heading inserted for this content.

Body: While the majority of the content in the body of this submission is interesting and makes a valuable contribution to the field I am left asking the following questions:

1. Where is pedagogy? In the abstract and the sentence at the end of the first paragraph this is mentioned- but is no expanded on? Nor positioned in the field- a number of researchers-academics are wrting and researching regarding new materialism and pedagogy.

2. Where is the author/s positioning regarding sound as matter? References etc

3. Need to consistently underpin and position, referencing is lacking. Can read as sweeping statements etc.

4. Structure of writing-content needs to be addressed- some very short sub-heading sections etc- this needs greater planning-editing to ensure there is a flow and building of an argument- link paragraphs and link sections. 

Written expression can be overly jargonistic and verbose- do not use seminal

Conclusion: Is very short needs further work and bringing together main ideas-premise etc.

Overall, I think this practice research is incredibly interesting but I need to hear the work- sonic intra-faces etc. links to sound files etc would really enrich the work. As would an entanglement of images-visuals and sound.

I indicated major changes in the overall judgement because I am constructing structure as holistic- the entire paper needs to be addressed-structurally- rather than just sections. However, I would encourage the author to address the above as this work-making can make a contribution to the field and has a lot of potential. 

Thank you. 

Author Response

Response to Reviewer (1) Comments

The premise for the paper is very interesting, hence my interest in reviewing this paper. The author/s state that the focus for the paper is unpicking a socially engaged art practice, Public Kitchen, through a new materialist lens and the intra-actions of sound and bodies and the agency-pedagogy of sound as matter. However, while the content is highly engaging, there are areas that in my professional judgement need to be addressed

Point 1:  Title: grabs the reader’s attention however, I question the use of interface specifically if you are using new materialist theories to underpin- could this be reworded to intraface?

Response 1: This has been changed and I agree that this makes the paper clearer and consistent with new materialist theories.

Point 2:  Abstract: Is overly jargonistic- sentences do not link, and the flow-coherence is  hampered. Need more straight forward communication regarding what, why, how, so what- and ensure sentences link-flow.

Response 2: I have re-written the whole abstract for your review.

Point 3:  Why start the paper with a quote and then not explain-expand-clarify. while this an academic writing genre convention- I find it does not add anything to the paper.

Response 3: Please refer to line 175.

Point 4: Introduction: The content after the sub-heading introduction is very looooong and is not functioning as an introduction to the premise-content-argument of the entire paper. I suggest the author/s need to write an introduction and include before this content with a new sub-heading inserted for this content.

Response 4: Please refer to line 62 with the sub-heading: Introduction..

Point 5: Body: While the majority of the content in the body of this submission is interesting and makes a valuable contribution to the field I am left asking the following questions:

Where is pedagogy? In the abstract and the sentence at the end of the first paragraph this is mentioned- but is no expanded on? Nor positioned in the field- a number of researchers-academics are writing and researching regarding new materialism and pedagogy.

Response 5: Please read the Introduction where I have made changes in the text to address pedagogy/authors and new materialism precedents/authors.

Point 6: Where is the author/s positioning regarding sound as matter? References etc.

Response 6: Line 155. Sound matter is therefore generated by making these conditions/forces a constant process of engagement, where thinking and acting ‘from the middle out’—drawing on what is intrinsic or embedded, creating ways of shifting into each other and attuning to these fields of difference. A Public Kitchen can, therefore, be seen to represent an active pedagogy for organizing and responding collectively to the local, through a spectrum of sound phenomena where home is a middling, while still operating as an independent nomadic event with many, and potentially on-going, transnational iterations.

I believe I am creating something new in terms of vital matter in sound performance and this is fully laid out in this manuscript. I am foremost a visual artist creating immersive sound performance sculptural installations and not a sound technician or sound theorist. What is crucial to me is to maintain a hybrid/intermedia practice that allows for collaboration in a transdisciplinary way.

Point 7: Need to consistently underpin and position, referencing is lacking. Can read as sweeping statements etc.

Response 7: I have lengthened this manuscript extensively now and it is now almost 15,000 words with the inclusion of footnotes. It started as a 6000 – 9000 word limit. In turn, I had to keep my statements short and succinct. I was told by the original editors not to contextualize everything since the title of the Journal on Feminist New Materialism already points to the reader that some of the definitions do not have to be contextualised since they are familiar with these terms. I have paraphrased on many occasions. Yes. Having said that, and in consideration of this peer review, I am honouring the comments and arguments in this manuscript as the feedback is crucial.

I have now mostly re-written the first half of this manuscript to enlighten the reader about the artwork and this practice-led research as a feminist new materialist. I hope this has helped immensely to fill in the gaps that you are alluding to in the manuscript. I have written a Preface that includes links to my personal website to enable the reader to first view the audio-visual documentation and listen to sound bites before or during the reading of this paper. Hopefully this will help the reader understand the theoretical approach to this manuscript and also the experience of live sound performance art which can never replace the live experience for the viewer.

Point 8: Structure of writing-content needs to be addressed- some very short sub-heading sections etc- this needs greater planning-editing to ensure there is a flow and building of an argument- link paragraphs and link sections.

Response 8: I have now inserted sub-heading sections also to build a flow and argument to the manuscript that links all sections.

Point 9: Written expression can be overly jargonistic and verbose- do not use seminal.

Response 9: I am interested in objective comparisons that can be made between kitchen tasks and live performance and what that can offer or contribute as a proposition that is actual through feeling and experience in the live act with all of its potential. I create an experiential event with that it is the action between (and not in-between) that matters with indeterminate human and nonhuman bodies and put this into practice; thereby, this manuscript is to be read as theory with practice as much as practice-led research.

Point 10: Conclusion: Is very short needs further work and bringing together main ideas-premise etc.

Response 10: I have lengthened the conclusion now.

Point 11:  Overall, I think this practice research is incredibly interesting but I need to hear the work- sonic intra-faces etc. links to sound files etc would really enrich the work. As would an entanglement of images-visuals and sound.

Response 11: This has all been adjusted and responded to in Point 7.

Point 12: I indicated major changes in the overall judgement because I am constructing structure as holistic- the entire paper needs to be addressed-structurally- rather than just sections. However, I would encourage the author to address the above as this work-making can make a contribution to the field and has a lot of potential.

Response 12:

I hope that I have answered all your questions and comments, suggestions. Thank you for your time and consideration in reading this manuscript and I hope you will take the opportunity to read my full PhD thesis dissertation from the Victorian College of the Arts on: Sonic Recipes from a Public Kitchen: Participatory Feminist Performance Art. It will be available shortly on the thesis repository at the University of Melbourne, Melbourne, Australia in 2019.

Thank you!

Juliana España Keller

Reviewer 2 Report

Peer review Humanities

This is an interesting piece that draws on appropriate references in current materialist feminist scholarship. I support its publication but recommend some revisions. Esxpecially the first third requires some conceptual sharpening and would benefit from some less jargon and more concrete description and explanation. Keep your reader in mind. Sometimes the argument gets lost in jargon. 

The “object” of analysis—“Public Kitchen” performances—should be described and contextualized explicitly and early on, before it is examined and refracted through post-Deleuzian terminology. (This might also require a minor reworking of the abstract). This might also entail contextualizing some of the secondary sources—what is the context of Barad’s, Braidotti’s conceptualizations. Readers not familiar with these references may be discouraged by the inflationary use of specialized concepts like intra-actions, entanglements, transversal flows, etc.

Especially the second part of the paper is strong in this regard, providing useful context. Please consider a careful review of punctuation (comma, semicolon, em-dash). Many references are unclear. Without page numbers, it is unclear whether the author is paraphrasing Barad and Braidotti, borrowing their argument, summarizing their ideas, etc. Maybe reconsider the use of the term “artwork” with respect to Public Kitchen performances (alternative: piece of performance art). 

Please find below some more detailed line-edits and comments.

The use of italics is unclear and inconsistent. (e.g. line 44, 55, 58, etc.), if these are quotes, they should be marked as such.

“The research generated by the artwork, a Public Kitchen, contributes to feminist new materialist discourse by making the notion of human-non-human agency graspable.” 

-> in references to “a Public Kitchen”, it remains unclear whether this is the title of a performance or a description

-> be consistent with italicizations (e.g. line 30/31)

“where the human and non-53 human fuse to take on a positive and affirmative character.”

-> this is unclear. what is a positive, affirmative character?

“Within each iteration, each participant is triggered by the vibrational sensation of sound that rises to the surface through the mind/body. Affect can be felt as sound that behaves as active matter—to listen and absorb sound material activated through the playing of kitchen objects, deeply. This triggers physical movement felt in the transmission within and between bodies.” 

-> this examination is quite vague. who is perceiving “the vibrational sensation of sound?” 

-> rather than immediately skipping into metaphor or analysis, it would be helpful to first describe the performance.

“this research seeks to address resonant frequency oscillating in sound that can be observed as vibrational bodies performing with the kitchen objects (reverberating as an extension of the self).” 

-> this remains very vague and jargonesque. What does it mean to “address resonance frequency oscillating in sound that can be observed as vibrational bodies”?

-> even if the authors are using the concept of resonance frequencies metaphorically, they need to be precise as to what they mean. Frequencies don’t oscillate. Air molecules oscillate in the transmission of sound. Frequencies characterize an oscillation (how many oscillations per second). 

“As Elaine Swan considers, there are colonial and anti-colonial dynamics at play between masculinity and femininity, specific ethnicities, multiculturalism and imperialism,”

-> requires clarification. Where are these dynamics at play? The authors need to be more specific. Otherwise this claim remains redundant.

“these dynamics are seen as post-human indicators from which we can interrogate, more closely, the connection (imagined or not) between food and the Other.”  

-> this remains too vague and fuzzy

line 79: the reference is not clear: Swan or Flowers?

“I would add that this agential relation is brought to the table as we struggle for coherence and continuity.” 

-> who is this “we”? 

-> coherence and continuity with respect to what?

line 90: unclear dash

“It is considered already embedded in these objects (the human-non-human), 91 spaces and things, as well as in spacetimemattering; it is diffractively working and conceptualizing difference through a spectrum of sound phenomena where home is a mutual relation of things and bodies inclusive of diverse participatory powers.”

-> specific terminology like Barad’s spacetimemattering or notion of diffraction require explanation.

line 99: unclear Whitehead reference

line 100: no semicolon

“Living always comes to terms with forms of dissonance emerging from a complex set of social 102 conditions, such as the auditory experience of sound that lacks musical quality, the sound that is a 103 disagreeable auditory experience as a form of noise. In turn, these conditions are in a constant process 104 of engagement, where thinking and acting ‘from the middle’—drawing on what is intrinsic or 105 embedded—creates ways of shifting into each other and attuning to these fields of difference.”

-> passages like these need to be more rigorously anchored in clear descriptions of the performances considered. Otherwise they remain too general and floating. 

line 115, this description would make sense at the very beginning. Up until this point, readers are only presented with a very vague image of what happens at a Public Kitchen event (actual cooking? rattling with kitchen utensils? audience participation? recitations? synthesizers?)

“Relationally, this research is a (re)imagining of the social, “

-> which research?

line159 “similarly-timed”—unclear

“rethinking the dualisms so central to our posthuman thinking. “

-> it seems these dualisms are more central to humanist thinking. 

-> who is “our” 

-> is there only one type of posthuman thinking?

Line 214 ff: this description would be helpful at the beginning of this paper.

line 220: tactile?

225: “particles entangling with particles in the quantum field of thinking; “

-> without explaining the specifics of Barad, this remains too jargonesque. Which part of this sentence paraphrases Barad? 

“materialist feminism considers how women and men of 255 various races and ethnicities are kept in their lower economic status due to a power imbalance that 256 privileges those who already have privilege, thereby protecting the status quo (Hennessy and Ingraham 257 1997).”

-> this doesn’t seem to be specific to materialist feminism. What distinguishes “materialist” feminism from other feminisms in this regard?

“The performing body engages in a transversal 268 connective experience of listening with the whole mind/body relationship in unison with other matter, 269 connecting to materials into an inequitable future and set of becomings (Braidotti 2002).”

-> this sentence includes a lot of specialized post-Deleuzian jargon. Maybe some of this can be weeded out, as it doesn’t add to the argument and remains vague in the absence of explanation.

line 271. This Michel de Certeau reference should be made at the first mention of “doing-cooking”

line 286: the concept of deep listening is interesting but nowhere properly defined. 

“Critical reaction to collaborative practices of social engagement is always problematic, “

-> unclear. Why?

line 361, insert page number Braidotti (this is a general tendency of this paper: to leave out page numbers. Please review)

line 430: cut “the”

line 446: rather keywords in music theory, or sound studies?

line 463: which part of the argument do the references refer to? unclear. 

line 471: delete “to”

“there is not one individual woman isolated by her own repressive environment, but many, in various forms of social class, locally and internationally. “

-> this statement is unclear. maybe highlight intersectional approaches. 

“Every Public Kitchen will seek to underscore these realities through the making of an electronic sound and performance artwork.” 

-> reconsider this generalization

“From one kitchen to the next, and through the agency of the tools, this research seeks to create an alternative economy through the power of the kitchen and the sound that engages these power relations.” 

-> what is this research? unclear.

“Noisy resonance oscillates at a vibration of larger amplitude”

-> avoid redundancy (resonance emerges through oscillations, rather than oscillating itself)

“A field of resonance is important to me in the material discursive and the transversal.”

-> explain what you mean by transversal (reference to Deleuze, Massumi, Reynolds?)

line 560: consider cutting “for me”

line 609: a woman’s  

line 621: “John Cage’s acceptance of noise and accident“ — could be even stronger: John Cage’s celebration of noise and accident

line 637: Maybe it’s worth considering whether fake news is a concept worth reifying, given that it emerges from alt-right rhetoric.

line 648: review commas and consider the insertion of another “as” after “distinguishes”

line 650: ethics flow

line 667: what is a quantum response? unclear, even metaphorically

line 668: cut one “are”?

lines 687-688: reconsider commas

Author Response

Response to Reviewer (2) Comments

This is an interesting piece that draws on appropriate references in current materialist feminist scholarship. I support its publication but recommend some revisions. Esxpecially the first third requires some conceptual sharpening and would benefit from some less jargon and more concrete description and explanation. Keep your reader in mind. Sometimes the argument gets lost in jargon.

The “object” of analysis—“Public Kitchen” performances—should be described and contextualized explicitly and early on, before it is examined and refracted through post-Deleuzian terminology. (This might also require a minor reworking of the abstract). This might also entail contextualizing some of the secondary sources—what is the context of Barad’s, Braidotti’s conceptualizations. Readers not familiar with these references may be discouraged by the inflationary use of specialized concepts like intra-actions, entanglements, transversal flows, etc.

Especially the second part of the paper is strong in this regard, providing useful context. Please consider a careful review of punctuation (comma, semicolon, em-dash). Many references are unclear. Without page numbers, it is unclear whether the author is paraphrasing Barad and Braidotti, borrowing their argument, summarizing their ideas, etc. Maybe reconsider the use of the term “artwork” with respect to Public Kitchen performances (alternative: piece of performance art).

Please find below some more detailed line-edits and comments.

Point 1: The use of italics is unclear and inconsistent. (e.g. line 44, 55, 58, etc.), if these are quotes, they should be marked as such.

Response 1: I have corrected the italics where necessary and only highlighted certain books or titles instead of key terms or phrases of importance in the text.

Point 2: “The research generated by the artwork, a Public Kitchen, contributes to feminist new materialist discourse by making the notion of human-non-human agency graspable.”

-> in references to “a Public Kitchen”, it remains unclear whether this is the title of a performance or a description

-> be consistent with italicizations (e.g. line 30/31)

Response 2: I have now mostly re-written the first half of this manuscript to enlighten the reader about the artwork and this practice-led research as a feminist new materialist. I hope this has helped immensely to fill in the gaps that you are alluding to in the manuscript. I have written a Preface that includes links to my personal website to enable the reader to first view the audio-visual documentation and listen to sound bites before or during the reading of this paper. Hopefully this will help the reader understand the theoretical approach to this manuscript and also the experience of live sound performance art which can never replace the live experience for the viewer.

Point 3: where the human and non-53 human fuse to take on a positive and affirmative character.”-> this is unclear. what is a positive, affirmative character?

Response 3:  Please refer to line 175.

Point 4: “Within each iteration, each participant is triggered by the vibrational sensation of sound that rises to the surface through the mind/body. Affect can be felt as sound that behaves as active matter—to listen and absorb sound material activated through the playing of kitchen objects, deeply. This triggers physical movement felt in the transmission within and between bodies.”

-> this examination is quite vague. who is perceiving “the vibrational sensation of sound?”

-> rather than immediately skipping into metaphor or analysis, it would be helpful to first describe the performance.

Response 4:  Please refer to the new introduction and preface.

Point 5: “this research seeks to address resonant frequency oscillating in sound that can be observed as vibrational bodies performing with the kitchen objects (reverberating as an extension of the self).”

-> this remains very vague and jargonesque. What does it mean to “address resonance frequency oscillating in sound that can be observed as vibrational bodies”?

-> even if the authors are using the concept of resonance frequencies metaphorically, they need to be precise as to what they mean. Frequencies don’t oscillate. Air molecules oscillate in the transmission of sound. Frequencies characterize an oscillation (how many oscillations per second).

Response 5: This has been revised and changed in the text. Please refer to the section on Resonance from lines 192 – 219. I have also made changes to the text where you are referring to resonant frequency starting on line 209.

Point 6:  As Elaine Swan considers, there are colonial and anti-colonial dynamics at play between masculinity and femininity, specific ethnicities, multiculturalism and imperialism,”

-> requires clarification. Where are these dynamics at play? The authors need to be more specific. Otherwise this claim remains redundant.

Response 6: Please refer to line 221 – 222 and the footnote (8) on “the other”.

Point 7:  “these dynamics are seen as post-human indicators from which we can interrogate, more closely, the connection (imagined or not) between food and the Other.”  

-> this remains too vague and fuzzy.

Response 7: Again, this is explained in footnote (8) in order to follow through to clarify this sentence.

Point 8: line 79: the reference is not clear: Swan or Flowers?

Response 8: This reference is co-authored.

Point 9: “I would add that this agential relation is brought to the table as we struggle for coherence and continuity.” -> who is this “we”?  -> coherence and continuity with respect to what?

Response 9: I have corrected this section and elaborated quite extensively.

Point 10: line 90: unclear dash.

Response 10: Corrected.

Point 11: “It is considered already embedded in these objects (the human-non-human), 91 spaces and things, as well as in spacetimemattering; it is diffractively working and conceptualizing difference through a spectrum of sound phenomena where home is a mutual relation of things and bodies inclusive of diverse participatory powers.”

-> specific terminology like Barad’s spacetimemattering or notion of diffraction require explanation.

Response 11: Diffraction (6) and spacetimemattering (12) as specific terminology have been inserted as footnotes now.

Point 12 and 13: line 99: unclear Whitehead reference

line 100: no semicolon

Response 12/13: Whitehead reference removed and line 100 corrected.

Point 14: “Living always comes to terms with forms of dissonance emerging from a complex set of social 102 conditions, such as the auditory experience of sound that lacks musical quality, the sound that is a 103 disagreeable auditory experience as a form of noise. In turn, these conditions are in a constant process 104 of engagement, where thinking and acting ‘from the middle’—drawing on what is intrinsic or 105 embedded—creates ways of shifting into each other and attuning to these fields of difference.”

-> passages like these need to be more rigorously anchored in clear descriptions of the performances considered. Otherwise they remain too general and floating.

Response 14: These concepts and ideas have been re-introduced and moved to line 153.

Point 15: line 115, this description would make sense at the very beginning. Up until this point, readers are only presented with a very vague image of what happens at a Public Kitchen event (actual cooking? rattling with kitchen utensils? audience participation? recitations? synthesizers?)

Response 15: Please refer to the new abstract, introduction and preface.

Point 16: “Relationally, this research is a (re)imagining of the social, “

-> which research?

Response 16: Addressed in the following subheading.

6. (Re)configuring our Relationship with the World in Co-becoming Processes

Point 17: line159 “similarly-timed”—unclear

Response 17: Replace by the word – creation.

Point 18: “rethinking the dualisms so central to our posthuman thinking. “

-> it seems these dualisms are more central to humanist thinking.

-> who is “our” -> is there only one type of posthuman thinking?

Response 18: Deleted the word – our.

Point 19: Line 214 ff: this description would be helpful at the beginning of this paper.

Response 19: Corrected.

Point 20: line 220: tactile?

Response 20: Replaced by the word – haptic.

Point 21: 225: “particles entangling with particles in the quantum field of thinking; “

-> without explaining the specifics of Barad, this remains too jargonesque. Which part of this sentence paraphrases Barad? 

Response 21: I have elaborated on this sentence and re-thought and added to this concept on line 386.

Point 22: materialist feminism considers how women and men of 255 various races and ethnicities are kept in their lower economic status due to a power imbalance that 256 privileges those who already have privilege, thereby protecting the status quo (Hennessy and Ingraham 257 1997).”

-> this doesn’t seem to be specific to materialist feminism. What distinguishes “materialist” feminism from other feminisms in this regard?

Response 22: I have worked on this much more to explain the differences and speak to feminism. Lines 420 – 428.

Point 23: “The performing body engages in a transversal 268 connective experience of listening with the whole mind/body relationship in unison with other matter, 269 connecting to materials into an inequitable future and set of becomings (Braidotti 2002).”

-> this sentence includes a lot of specialized post-Deleuzian jargon. Maybe some of this can be weeded out, as it doesn’t add to the argument and remains vague in the absence of explanation.

Response 23: I have removed this sentence.

Point 24: line 271. This Michel de Certeau reference should be made at the first mention of “doing-cooking”.

Response 24: This has been corrected with foot-note (1).

Point 25: line 286: the concept of deep listening is interesting but nowhere properly defined.

Response 25: Lines 456 – 465 have now elaborated on the concept of deep listening.

Point 26: “Critical reaction to collaborative practices of social engagement is always problematic, “ -> unclear. Why?

Response 26: Please see Line 515.

Point 27: line 361, insert page number Braidotti (this is a general tendency of this paper: to leave out page numbers. Please review)/

Response 27: Done.

Point 28 and 29: line 430: cut “the”

line 446: rather keywords in music theory, or sound studies?

Responses to 28 and 29: Corrected.

Point 30: line 463: which part of the argument do the references refer to? unclear.

Response 30: Corrected. Lines 645 – 646.

Point 31: delete “to”.

Response 31: Corrected.

Point 32: “there is not one individual woman isolated by her own repressive environment, but many, in various forms of social class, locally and internationally. “

-> this statement is unclear. maybe highlight intersectional approaches.

Response 32: Corrected and explained in lines 673 – 679.

Point 33: “Every Public Kitchen will seek to underscore these realities through the making of an electronic sound and performance artwork.”

-> reconsider this generalization

Response 33: Corrected in lines 687 – 689.

Point 34: “From one kitchen to the next, and through the agency of the tools, this research seeks to create an alternative economy through the power of the kitchen and the sound that engages these power relations.”

-> what is this research? unclear.

Response 34: This idea is engaged in lines 687 – 689.

Point 35: “Noisy resonance oscillates at a vibration of larger amplitude”

-> avoid redundancy (resonance emerges through oscillations, rather than oscillating itself).

Response 35: Corrected.

Point 36: “A field of resonance is important to me in the material discursive and the transversal.”

-> explain what you mean by transversal (reference to Deleuze, Massumi, Reynolds?)

Response 36: I have inserted a footnote to explain the word in footnote (23).

Points 37 and 38: line 560: consider cutting “for me”

line 609: a woman’s 

Response 37 and 38: Corrected.

Point 39: line 621: “John Cage’s acceptance of noise and accident“ — could be even stronger: John Cage’s celebration of noise and accident

Response 39: Corrected. Well stated!

Point 40: Maybe it’s worth considering whether fake news is a concept worth reifying, given that it emerges from alt-right rhetoric.

Response 40: Corrected but not omitted in line 821.

Point 41 and 42: line 648: review commas and consider the insertion of another “as” after “distinguishes”. line 650: ethics flow

Response 41 and 42: Corrected.

Point 43: line 667: what is a quantum response? unclear, even metaphorically.

Response 43: Responded to on line 851 with a footnote on Barad. Footnote (28).

Point 44 and 45: line 668: cut one “are”? lines 687-688: reconsider commas.

Response 44 and 45: Corrected.

Appendix:

I hope that I have answered all your questions and comments, suggestions. Thank you for your time and consideration in reading this manuscript and I hope you will take the opportunity to read my full PhD thesis dissertation from the Victorian College of the Arts on: Sonic Recipes from a Public Kitchen: Participatory Feminist Performance Art. It will be available shortly on the thesis repository at the University of Melbourne, Melbourne, Australia in 2019.

Thank you!

Juliana España Keller

This manuscript is a resubmission of an earlier submission. The following is a list of the peer review reports and author responses from that submission.